# Reinforcement Learning-based Layer-wise Aggregation for Personalized Federated Learning

## Abstract

A key challenge in Federated Learning (FL) is statistical heterogeneity, which may result in slow convergence and accuracy reduction. To tackle this problem, personalized federated learning (PFL) aims to adapt the global model to the individual data distribution of each client. One approach for this is personalized aggregation, which automatically determines how much each client can benefit from other clients' models. This paper proposes a new PFL method based on two principles: a) shared knowledge and personalized knowledge are reflected in different layers of the network and b) clients with more data should contribute more to shared knowledge, while knowledge transfer from similar clients can boost personalization. Based on these, we propose a reinforcement learning-based layer-wise aggregation method (pFedRLLA) that applies different mechanisms for different neural network layers. For layers representing shared knowledge, aggregation is carried out based on the size of the local data samples of the client. For layers representing personalized knowledge, a deep reinforcement learning (DRL) agent is used to generate personalized aggregation weights. To ascertain efficiency and scalability, we train a single DRL agent (for all users) that operates on the server-side and takes as input a subset of user models. To further reduce its state-space, we design a multi-head auto-encoder to obtain low-dimensional embeddings of user models. Extensive experiments on benchmark datasets for variable data heterogeneity levels reveal that the proposed algorithm consistently outperforms baselines in terms of both higher accuracy (up to +3.1%) and faster convergence (a reduction of global rounds by up to 20.5%).

## 1 Introduction

Artificial Internet of Things (AIoT) is the domain where a large number of edge devices (such as mobile phones, sensors, cameras, etc.) generate big volumes of data that are used to train machine learning models in order to provide intelligent services. To this end, traditional centralized machine learning methods (LeCun et al., 2015; Dean et al., 2012) face many challenges, such as excessive communication burden, device battery consumption, and threats to data privacy and security. In order to overcome these problems, Federated Learning (McMahan et al., 2017) emerged as a new paradigm for distributed learning (Bottou, 2010), tailored to address the aforementioned challenges. FL involves four operations: 1) edge devices download the global model, 2) they fine-tune it based on local training, 3) they upload the updated model, and 4) the server carries model aggregation.

In FL, a core issue is statistical heterogeneity, i.e., the situation where the data of different clients are unbalanced in volume and not independent and identically distributed (non-IID). This makes it difficult to obtain a shared global model that generalizes to all clients–for example, due to client drift (the case where the model overfits to a user's data)–and slows down the progress of the FL process.

Personalized Federated Learning (PFL) is a framework that aims to generate customized models for clients based on their local data while benefiting from collaborative training. In contrast to classical FL which trains a common model for all users (e.g., (McMahan et al., 2017; Wang et al., 2022; Gong et al., 2022)), PFL considers the heterogeneity and diversity of clients' data and tasks. PFL methods can be classified into two categories: (i) **Global model personalization**, where the users personalize the shared model: there are data-driven methods (Wang et al., 2020; Chai et al., 2020;

Duan et al., 2021) that attempt to reduce the distribution imbalance among clients, as well as model-based methods (Li et al., 2021a; Finn et al., 2017; Yao & Sun, 2020; Li et al., 2020) that optimize the global FL model for personalized needs, using techniques such as regularization (Li et al., 2020; 2021b; T. Dinh et al., 2020), meta-learning (Finn et al., 2017; Fallah et al., 2020), and transfer learning (Li & Wang, 2019). (ii) **Client-specific model generation**, which trains a personalized model for each client: this includes structure-based (Liang et al., 2020; Collins et al., 2021; Oh et al., 2021) and similarity-based methods (Smith et al., 2017; Huang et al., 2021). Notably, *personalized aggregation* is a popular approach (Zhang et al., 2022; 2020; Ma et al., 2022).

In this paper, we mainly focus on image classification tasks, for which Convolutional Neural Networks (CNNs) are used. CNNs consist of two main components: a feature extractor and a classifier. In representation learning (Kang et al., 2019; Yu et al., 2020), all tasks share the same feature extractor, while each task has its own classifier. Kornblith et al. (2019) verified that increasing the amount of data significantly improves the performance of the model on various tasks, while the connection with the label distribution of the tasks is relatively weak. Drawing inspiration from this, we divide the CNN model for each client into two parts: the first part (the body) is responsible for learning domain representation, and the second part (the head) is task-specific. Each client maintains a personal model (both for body and head), but we adopt different aggregation strategies for the two parts. The body is more dependent on data quantity, so we use weighted aggregation based on the data sizes. The head is more susceptible to the distribution of client data. For this reason, we design a DRL algorithm that generates personalized aggregation weights by learning the potential correlations between clients. The state of the DRL agent consists of the personalized models of clients selected in the previous round. The reason for not choosing all clients as the state is that FL may involve a large number of clients (e.g., thousands), so this would result in a large state space which, in turn, would cause the DRL training to be slow and expensive. Extensive experiments show that our design is effective even when the clients selected in the previous round are substantially different in terms of distribution with the currently selected clients (in such case, the weights automatically detect and capture this: the self-weight will be large and the unrelated clients will contribute little). Furthermore, in view of the high dimension of the model parameters (for example, a 6-layer CNN has 0.925M parameters), we design a multi-head auto-encoder (MHAE) to obtain low-dimensional embeddings of the heads. Unlike most FL methods, pFedRLLA no longer generates a global model. Instead, it produces a personalized model for each client, which is more tailored to individual needs than methods in category (Global model personalization). Besides, we take into account inter-client relationships when generating personalized aggregation weights for client $i$, unlike methods such as pFedLA that only consider the connection between the global model and client $i$.

**Contributions:**

- We propose a personalized aggregation method that generates a personal model for each client. Our method utilizes a layer-wise approach, adopting different aggregation strategies for different neural network layers. This ensures that the resulting model has a universally effective feature extractor and a classifier tailored to individual needs.

- We design a DRL agent to achieve the aggregation of the head parameters for every client. Its training is carried using a compound reward, which takes into account both the improvement of validation accuracy as well as the similarity between clients. To further expedite the process, we design a multi-head auto-encoder that reduces the state space. Besides, the aggregation of the bodies is carried using weighted aggregation based on the data volumes of the participants.

- We conducted numerous experiments on four datasets. Our proposed solution outperforms other baselines in all settings (3.13% higher accuracy on CIFAR-100) and achieves a faster convergence (a maximum 20.4% of communication savings over the best performing baseline on CIFAR-10).

The novelty of this paper does not lie in the tools used (i.e., RL, MHAE) but rather in the succinct design and implementation of a data-driven solution for PFL that is shown to be effective and robust via extensive experimentation in diverse scenarios. To that end, there are two main distinctions with existing methods. First, the use of a *single* DRL agent at the server side which allows for scalability. The efficacy of this choice is made possible by three means: a) pre-training based on a compound reward that balances accuracy, and model similarities, b) the setting of single-step task,

which provides a more pronounced reward signal and simplifies the collection of training data and c) the use of MHAE, that reduces the complexity and captures the similarities more effectively. Second, we have adopted the simplest possible layer-wise approach (body and head), which allows to obtain feature extractor and a more customized classifier efficiently.

## 2 RELATED WORK

Personalized Federated Learning (PFL) intends to produce customized models that meet users' demands by capitalizing on information exchanges through the federated process in addition to local data. As mentioned earlier, PFL algorithms can be divided into two categories.

**Global model personalization.** FedDRL (Nguyen et al., 2023) employs a DRL approach to derive aggregation weights for the global model. In model-based methods, the use of regularization is popular. FedProx (Li et al., 2020) adds a penalty term in the local loss function of the client to capture the dissimilarity between the global and local models. Per-FedAvg (Fallah et al., 2020) and pFedMe (T. Dinh et al., 2020) are methods that adopt meta-learning. The former is a variant of FedAvg built on top of the MAML framework (Finn et al., 2017) that targets to compute a shared model that new users can easily adapt to with minimal local fine-tuning. pFedMe uses the Moreau envelope as a regularizer and optimizes the personalized model and the global model simultaneously. Nevertheless, and in contrast to our proposed solution, these methods rely on training a single global model and are not well suited to address personalization in scenarios where there are significant differences among client data distributions.

**Client-specific model generation.** There is a multitude of relevant methods such as representation learning-based, cluster-based, and personalized aggregation-based methods. FedRep (Collins et al., 2021) and FedBABU (Oh et al., 2021) were designed based on representation learning, i.e., they decouple the model parameters into two parts: base layers and personalized layers. The former are obtained from the aggregated model held by the server while the latter are updated through local training. Clustered federated learning (Ghosh et al., 2020; Long et al., 2020) uses similarity scores to divide clients into different groups and maintains one model per group at the server. To further enhance personalization, there are methods that generate customized models for each client through personalized aggregation, an approach that we also adopt in this paper. For instance, FedAMP (Huang et al., 2021) utilizes a message-passing mechanism to perform personalized aggregation among similar clients, while FedALA (Zhang et al., 2022) built an Adaptive Local Aggregation (ALA) module to incorporate knowledge from the global model into the local model.

Our proposed solution lies in the second category, and specifically invokes representation learning (Bengio et al., 2013). The main novelty lies in the design of different aggregation strategies for different parts of the neural network: the aggregation of the body allows us to extract better representations from data-rich clients, while the aggregation of the head leverages task-similar clients to personalize the model to better meet the unique needs of each client. The main differences with related baselines are as follows. pFedLHN Zhu et al. (2023) employs layer-wise modules with parameter-sharing mechanisms in the hypernetwork to create personalized models for each client. The hypernetwork generates these personalized models by utilizing layer embeddings uploaded by the clients themselves, and does not rely on learning from other clients. Moreover, uploading these layer embeddings to the server introduces additional communication. FedPAC (Xu et al., 2023) adopts local-global feature alignment to learn a better representation and formulates an optimization problem to estimate optimal aggregation weights for the classifier. However, this method introduces additional communication overhead and client-side computation burden. Furthermore, its applicability is limited to scenarios involving label distribution shift. FedALA adaptively aggregates the global model and the local model on the client side to initialize the local training. Therefore, the performance has a high requirement on the quality of the global model, otherwise, a poor initialization may stagnate the process. The most similar to our algorithm is pFedLA. It maintains a hypernetwork for each client on the server, which can generate personalized aggregation weights for each layer of the corresponding client model. However, this method requires training individual networks for each client on the server side, therefore, it suffers from limited scalability. In contrast, our method only needs a single hypernetwork for all clients. Besides, in our design, the DRL agent takes as input models corresponding to participants in the previous round and effectively identifies similar clients, in contrast to pFedLA. Finally, our method does not only consider user similarity but also the amount of local data, which have higher impact on the body of the network.

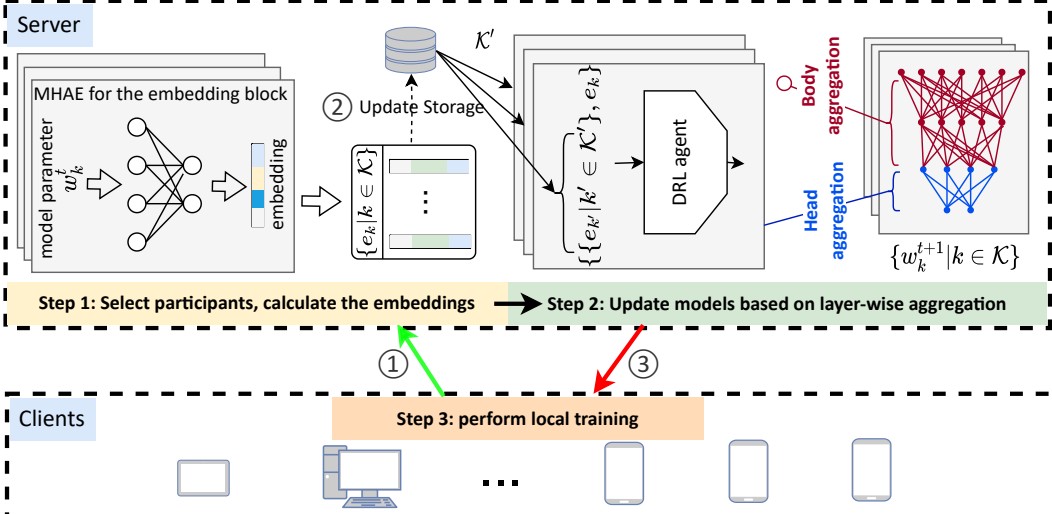

Figure 1: The overall architecture of pFedRLLA. The algorithm iterates through three main steps: In Step 1, the server selects $k$ participants. These upload their models as in ①. The server computes the embeddings of the models through a multi-head-autoencoder (MHAE). In Step 2, the server uses layer-wise aggregation for each selected client using the information held in storage (for the body the weights are proportional to the data sizes of the clients, while for the heads they are calculated by the DRL agent). Finally, the server updates the storage with the embeddings and data sizes pertaining to the selected clients ②. In Step 3, the selected users download their corresponding updated models (③) and perform local training.

## 3 ARCHITECTURE OF PFEDRLLA

The architecture of our proposed solution is illustrated in Fig. 1 and the detailed description is given in Alg. 1. On the server side, we maintain a DRL agent, which generates aggregate weights for all clients, and an MHAE, which reduces the dimensionality of model parameters. The details of their design will be detailed in the next section. At the beginning, all $N$ eligible devices download the initial random model from the server and train locally to initialize their local models. After that, the algorithm iteratively computes personalized models as follows.

**Step 1:** In round $t$, the server randomly selects a subset of clients. These clients upload their local models, and then the server extracts the embeddings of their models through a multi-head-autoencoder (MHAE).

**Step 2:** The server applies a layer-wise aggregation for each selected client. For the body, aggregation is based on the volume of the local data size (lines 1-2 in Alg. 2). For the head, the server combines its embedding with the embeddings of the clients selected in the previous round, and the DRL agent is invoked to generate the aggregation weights (lines 3-5 in Alg. 2). Finally, the server updates the storage with the embeddings and data sizes pertaining to the selected clients (to be used at the next round).

**Step 3:** The selected clients download their corresponding updated models and perform local training (lines 10-13 in Alg. 1).

Note that this procedure incurs no extra computational or communication burdens on the client side. Furthermore, it does not impact the overall training time, because both the MHAE and the DRL agent are pre-trained (detailed in Appendix D), and are further fine-tuned during the PFL process (detailed in Appendix E). Finally, scalability at the server side is ensured by controlling the size of the subset of participants (i.e., the store does not need to maintain information for all users).

### 3.1 DIMENSION REDUCTION

A typical deep learning model comprises of layers that may vary in size. In order to deal with the large dimension, several FL algorithms (Wang et al., 2020; Ghosh et al., 2020) concatenate different

**Algorithm 1** pFedRLLA

**Parameter**: $N$ clients, Selection ratio $c$, Communication rounds $T$.

1: **for** each round $t = 1$ **to** $T$ **do**
2:    $\mathcal{K} \leftarrow$ (random set of $\lceil c \cdot N \rceil$ clients)
3:    Selected users **upload** their models
4:    **Server** computes the embeddings:
      $e_k = MHAE(w_k), k \in \mathcal{K}$
5:    **for** client $k \in \mathcal{K}$ **do**
6:       $w_k \leftarrow$ Aggregation$(k, \mathcal{K}')$
      (update models, as detailed in Alg. 2.)
7:    **end for**
8:    **Server** updates the embeddings in storage;
9:    $\mathcal{K}' = \mathcal{K}$ ▷ record previously selected clients [a]

10:    **for** client $k \in \mathcal{K}$ **in parallel do**
11:       **Client downloads** $w_k$ from server
12:       Conduct local training
13:    **end for**
14: **end for**

---

[a]For the first round, $\mathcal{K}'$ is the set of currently selected clients.

**Algorithm 2** Layer-wise aggregation

Aggregation$(k, \mathcal{K}')$        ▷ Run on **server**
**Parameter**: $\Psi$ is the model's body, $\Phi$ is the model's head, $d_i$ represents the size of client $i$'s local dataset.
**Body aggregation:**
1:    Calculate weights based on local data sizes:
      $d \leftarrow d_k + \sum_{k' \in \mathcal{K}'} d_{k'}$
      $\rho \leftarrow \{\{\rho_{k'} := \frac{d_{k'}}{d} | k' \in \mathcal{K}'\}, \rho_k := \frac{d_k}{d}\}$
2:    Update the body as:
      $\Psi_k \leftarrow \rho_k \Psi_k + \sum_{k' \in \mathcal{K}'} \rho_{k'} \Psi_{k'}$
**Head aggregation:**
3:    Use the embeddings
      $e_k = MHAE(\Phi_k)$
      $e_{k'} = MHAE(\Phi_{k'}), k' \in \mathcal{K}'$
      $S_k = \{\{e_{k'} | k' \in \mathcal{K}'\}, e_k\}$
4:    Calculate the aggregation weight $A_k$
      $A_k \leftarrow DRL(S_k)$;
      $A_k := \{\{p_{k'} | k' \in \mathcal{K}'\}, p_k\}$
5:    Update the head as:
      $\Phi_k \leftarrow p_k \Phi_k + \sum_{k' \in \mathcal{K}'} p_{k'} \Phi_{k'}$
6: return $\{\Psi_k, \Phi_k\}$

layers into a large tensor and then invoke dimension reduction, e.g., using Principal Component Analysis (PCA). Nonetheless, PCA overlooks the structural features of the network. For this reason, we opt to devise a multi-head auto-encoder (MHAE) for dimension reduction. In this, the multi-head architecture aims to transform the task heads into feature matrices of uniform dimensions (using PAC). Then, we employ an LSTM auto-encoder to learn the correlations between these feature matrices. The comparison of dimensionality reduction results between MHAE and PCA is shown in Fig. 6 in the Appendix.

### 3.2 LAYER-WISE AGGREGATION

Our proposed layer-wise aggregation algorithm adopts different strategies for different parts to achieve model personalization. More specifically, the network is divided into two parts: the body, that is responsible for feature extraction, and the head, that corresponds to responsible for the decision layers of the CNN. For the feature extraction layers, previous works (Collins et al., 2021; Oh et al., 2021) have shown that heterogeneous data distributed across tasks may share a common representation despite having different labels. This motivates the aggregation of feature extraction layers that ignore the local label distribution and only focus on the data volume. Specifically, for body aggregation, we use weighted averaging based on the data size of each client. In contrast, the classifier layers (i.e., heads) are closely tied to the tasks handled by the clients and are therefore more influenced by the local data distributions. Therefore, we propose a DRL-based approach to generate personalized weights. The detailed process of the head aggregation is presented in the following subsection.

#### 3.2.1 HEAD AGGREGATION

We employed DDPG (Lillicrap et al., 2015) (deep deterministic policy gradient), leveraging its actor-critic architecture. The actor network is tasked with acquiring the aggregation weights, while the critic network estimates the value function associated with the current action, thereby guiding the actor's subsequent actions. The choice of the DDPG algorithm stems from two primary reasons: first, our task operates within continuous state and action spaces, and second, DDPG enables the generation of deterministic policies. The task of learning aggregation weight is formulated as a single-step process where the generation of one set of aggregation weights (for each participating client based on the embeddings at the previous round) constitutes an episode (more details are provided in the Appendix C). One-step tasks typically feature a more pronounced reward signal, facilitating the agent's understanding of when to be rewarded or penalized. The definitions of State, Action, and Reward are detailed next.

*State:* The state of client $k$ contains the embeddings of the model parameters of the participants in the previous round $\{e_{k'}|k' \in \mathcal{K}'\}$ as well as the embedding of its own model $e_k$ (line 3 of Alg. 2).

*Action:* The action is the set of weights corresponding to the previous participants and client $k$ itself, $A_k = \{\{p_{k'}|k' \in \mathcal{K}'\}, p_k\}$, the sum of $A_k$ equals 1, which is achieved by adding a softmax activation function to the final layer of the actor network.

*Reward:* The reward for a single-step is computed as a compound of three sub-rewards with weights $\{\beta_i\}_{i=1}^3$, which are set as $\{1, 1, 2\}$ in our experiments (see Appendix F.1).

$$R = \sum_{i=1}^3 \beta_i r_i, \tag{1a}$$

$$r_1 = e^{\text{acc}_k - \text{acc}_{\text{tar}}} - 1, \tag{1b}$$

$$r_2 = \text{acc}_k^{\text{agg}} - \text{acc}_k^{\text{pre}}, \tag{1c}$$

$$r_3 = -\|A_k - V_k\|^2. \tag{1d}$$

In (1b), $\text{acc}_k$ is the test accuracy achieved by the model of client $k$ on the held-out validation set, while $\text{acc}_{\text{tar}}$ is a predetermined target accuracy. This can be adjusted with the process of training. This term is negative unless the user has met its local needs.

The second term (1c) rewards improvement in test accuracy after aggregation: $\text{acc}_k^{\text{agg}}$ denotes the test accuracy of the newly aggregated model of client $k$ on the validation set, and $\text{acc}_k^{\text{pre}}$ represents the accuracy of client $k$ before updating its model by layer-wise aggregation.

The last term (1d) is used to generate weights to identify model similarities (this term is negative since it penalizes the discrepancy). In specific, $A_k$ denotes the normalized weights (line 4 of Alg. 2) generated by the DRL agent and $V_k$ is given by:

$$V_k = \{\{d(e_k, e_{k'})|k' \in \mathcal{K}'\}, 1\} / \left( \sum_{k' \in \mathcal{K}'} d(e_k, e_{k'}) + 1 \right), \tag{2}$$
$$\text{where} \quad d(e, e') := \frac{\cos(e, e') + 1}{2}.$$

The cosine similarity is denoted by $\cos(\cdot, \cdot)$ (note that $1 = d(e_k, e_k)$ in $V_k$ reflects the self-similarity), so that $V_K$ corresponds to normalized cosine similarity.

## 4 EXPERIMENTS

In order to support the merits of our proposed algorithm, we have conducted a wide range of experiments on real datasets under variable levels of heterogeneity and in comparison with a dozen baseline methods. Our main findings enlist: (1) pFedRLLA features higher accuracy at the user end as well as a faster convergence rate. (2) Our ablation experiments provide empirical evidence supporting the validity of our design choices, specifically the utilization of weighted aggregation for body and personalized aggregation for head.

### 4.1 EXPERIMENTAL SETUP

We have used four datasets to evaluate pFedRLLA and other baselines, namely CIFAR-10, CIFAR-100, MNIST*, and Omniglot. For the first two datasets, we consider various levels of heterogeneity (see **Data Partioning** for the detailed description) based on a) a 'pathological' scenario, where each user contains only a small number of labels and b) a Dirichlet distribution with adjustable parameter. The latter two datasets are different from the aforementioned heterogeneous scenarios: the Omniglot dataset is gathered from 20 distinct users and is thus naturally partitioned into 20 FL clients. The MNIST* dataset is a hybrid dataset constructed from three datasets (MNIST, MNIST-M, and USPS), and in this case we established three distinct client groups, with each group containing different domain samples. We divide these datasets into training set, validation set, and test set with a ratio of 65:15:20. More details about the datasets are provided in Appendix B.1.

In order to test our method and all baselines, we use a 6-layer CNN architecture, as employed in FedALA (Zhang et al., 2022), for all four tasks. The learning rates of the local SGD solver are chosen from the candidate set $\{0.005, 0.01, 0.1, 0.2\}$ for best performance. Our experiments for

the CIFAR and MNIST* datasets consisted of 100 users, with 10% of them selected for each round (line 2 of Alg. 1). The Omniglot dataset contains data from 20 users and for this case, $c = 20\%$ is chosen. The reported accuracy reflects the average test accuracy across all users.

**Data Partitioning**. We adopted two data partitioning methods. (1) Pathological non-IID as in FedAvg (McMahan et al., 2017): first, we group all the data according to the label and then divide them into $s * N$ shards, where $s$ is the number of labels per client and $N$ is the total number of clients. Smaller values of $s$ indicate fewer data labels per client and a smaller number of clients with the same data label, resulting in a higher level of heterogeneity. However, in such case, the clients have more data for the corresponding classes (thus a larger $s$ is more challenging for this reason). (2) Dirichlet non-IID (Hsu et al., 2019): we use a vector $x_k$ to represent the probability distribution of class $k$ on different clients, which is sampled from the Dirichlet distribution, denoted as $\text{Dir}(\alpha)$; the smaller $\alpha$ is, the higher the level of heterogeneity.

**Baselines**. We compare pFedRLLA versus several state-of-the-art methods. The baselines include the traditional federated averaging and PFL methods from the two categories discussed in Section 2 (global model personalization and client-specific model generation). In the former category, we choose Ditto, APFL, FedRep, and FedBABU as baselines. In the latter, FedAMP, FedFomo, Fed-PHP, FedPAC, and FedALA are chosen. Last, we further include the variant of FedAvg where clients fine-tune the global model so as to personalize (listed as FedAvg+FT) and Local-only refers to when clients train locally without communicating with the server.

**Implementation**. We simulate all clients and the server on a workstation with 2 Intel® Xeon® 353 Gold 6330 CPUs and 8 NVIDIA® 354 GeForce RTX 3090 GPUs. All methods are implemented in PyTorch. Our code is available at: anonymous.4open.science/r/pFedRLLA-7BEF.

Table 1: Average test accuracy on CIFAR-10, CIFAR-100, Omniglot, and MNIST*; the selection ratio was set to 0.1 in all cases (except Omniglot, where it equals to 0.2). The first two datasets exhibit varying levels of heterogeneity, while the latter two datasets have a fixed level of heterogeneity (as per their construction). pFedRLLA improves the average accuracy by 1%-3%. The best results are plotted in bold and the second best are underlined.

| | Pathological non-IID | | | | Dirichlet non-IID | | | - | |
| | CIFAR-10 | | CIFAR-100 | | CIFAR-10 | | CIFAR-100 | Omniglot | MNIST* |
| | $s = 2$ | $s = 5$ | $s = 5$ | $s = 20$ | $\alpha = 0.1$ | $\alpha = 1$ | $\alpha = 0.1$ | - | - |
| Local-only | 82.88 | 61.17 | 65.96 | 32.36 | 84.58 | 49.76 | 39.55 | 15.92 | 34.27 |
| FedAvg | 86.26 | 79.43 | 70.14 | 46.20 | 84.31 | 69.05 | 49.77 | 44.89 | 72.27 |
| FedAvg+FT | 89.61 | 80.26 | 72.94 | 47.62 | 89.24 | 71.64 | 52.86 | 44.73 | 73.88 |
| Ditto | 89.40 | 80.00 | 72.36 | 47.49 | 86.81 | 70.78 | 53.21 | 45.36 | 70.46 |
| APFL | 89.43 | 80.10 | 72.82 | 48.01 | 89.38 | 71.41 | 52.76 | 44.80 | 73.77 |
| FedRep | 88.44 | 75.63 | 70.26 | 39.89 | 86.94 | 56.76 | 41.35 | 27.35 | 69.77 |
| FedBABU | 89.80 | 79.17 | 73.62 | 48.03 | 89.54 | 70.89 | 53.06 | 44.12 | 72.58 |
| FedAMP | 83.47 | 64.78 | 67.67 | 34.50 | 84.73 | 50.09 | 38.99 | 16.18 | 64.75 |
| FedFomo | 88.34 | 76.80 | 71.01 | 43.94 | 84.93 | 55.42 | 38.18 | 39.60 | 68.77 |
| FedPHP | 88.28 | 79.67 | 71.66 | 46.70 | 88.22 | 69.14 | 50.23 | 45.26 | 72.66 |
| FedPAC | 74.97 | 75.27 | 58.98 | 48.04 | 82.82 | 65.13 | 45.94 | 39.64 | 68.59 |
| FedALA | 89.38 | 79.04 | 70.37 | 44.22 | 87.94 | 69.47 | 44.50 | 45.20 | 73.58 |
| Ours | **90.81** | **81.76** | **76.06** | **51.15** | **90.26** | **72.46** | **53.79** | **48.43** | **74.58** |
| Impr. | 1.01 | 1.50 | 2.44 | 3.11 | 0.72 | 0.82 | 0.58 | 3.07 | 0.70 |

## 4.2 EXPERIMENTAL RESULTS

Table 1 exposes the test accuracy (averaged across all clients) achieved by each method after 500 global rounds (after which all listed methods have little or no accuracy gain with additional training). As it can be inspected, pFedRLLA consistently achieves higher accuracy than all tested baselines with the main advantages discussed in the following.

For the pathological non-IID case, a larger value of $s$ reflects a more complex classification task. This is because in such case the users have more labels (equal to $s$) but fewer data points per label

(equal to $\frac{D}{s*N}$; $N$ is the number of clients, $D$ is the dataset size). Therefore, in our experiments, the performance of each baseline decreases as $s$ increases. Besides, when comparing CIFAR-10 $s$ = 5 with CIFAR-100 $s$ = 5, the latter is a more heterogeneous scenario (in both cases there are 5 classes per client, but in the case of CIFAR-10 the 100 clients will be divided into 2 categories while there 20 categories in CIFAR-100). In Table 1, we observe that the improvement of pFedRLLA over the second best baseline increases from 1.5% to 2.4% from CIFAR-10 $s$ = 5 to CIFAR-100 $s$ = 5, which indicates a higher improvement for our algorithm in scenarios with higher heterogeneity. Last but not least, due to the characteristics of the data partitioning method, each client possesses only a specific subset of labels controlled by $s$, and these clients are grouped together with significant between-group variability and high within-group similarity. This characteristic allows our algorithm to generate better aggregation weights and produce more personalized models. In all four scenarios, our algorithm demonstrated a clear advantage, with a 3.11% lead in the best case (CIFAR-100 $s$ = 20) and a 1.01% improvement in the worst case (CIFAR-10 $s$ = 2).

In the Dirichlet non-IID scenario, the degree of improvement is comparatively less significant. This occurs because this setting raises considerable variability across clients' data in terms of both volumes as well as the distribution of labels. Therefore, the clients are less similar to one another in terms of data distribution (Fig. 7 in the Appendix), thus reducing the apparent benefits of collaboration in compromising of the resulting accuracy. Nevertheless, pFedRLLA consistently surpasses the baseline methods in accuracy.

Finally, we used two more heterogeneous scenarios: on the Omniglot dataset, we outperform the best performing baseline by 3.07%, while on MNIST*, our algorithm achieves a 0.7% improvement.

**Ablation studies.** Our algorithm adopts a layer-wise aggregation method (applying different aggregation strategies for the body and head), so ablation experiments are conducted to verify the effect of each mechanism. Table 2 presents the initial and personalized accuracy of each sub-variants in the three heterogeneous scenarios of CIFAR-100. We observe that both aggregation strategies contributed to accuracy improvement. In the IID case ($s = 100$), the advantage brought by the 'Body' strategy (weighted aggregation) is more pronounced. As heterogeneity increases (the smaller $s$ is, the higher the heterogeneity), the advantage of the 'Head' strategy (DRL aggregation) becomes more prominent, indicating a close relationship between the head and personalization in heterogeneous data settings. pFedRLLA utilizes both aggregation methods, resulting in better performance than all variants (i.e., that adopt one of the two strategies and use the simple aggregation strategy of FedAvg for the other). For comparison, we also added the setting aggregating the full model based on the DRL agent (which is trained for this setting), instead of just the heads. This is listed as 'DRL' in Table 2, and we can observe that the accuracy is noticeably inferior to our proposed solution that adopts the layer-wise aggregation strategy. Furthermore, we noted that the personalized accuracy of 'None' and 'Body' are comparable in $s$ = 10, which is expected as the latter merely introduces a simple modification (i.e., size-proprotional weights) to the body aggregation. However, it is worth noting that the 'Body' strategy consistently exhibits better initial accuracy, potentially attributed to its consideration of local hard-won models, enabling the newly aggregated model to extract representations that align better with the characteristics of the local dataset.

Table 2: Ablation study. The table displays the before-personalization (BP) and after-personalization (AP) accuracy results from five different algorithms. The distinction among the five settings lies in which part of the neural network adopts our proposed aggregation methods, while the remaining parts employ simple averaging (as in FedAvg).

|  | None | | Body | | Head | | Both | | DRL | |
| --- | --- | --- | --- | --- | --- | --- | --- | --- | --- | --- |
|  | BP | AP | BP | AP | BP | AP | BP | AP | BP | AP |
| $s = 10$ | 0.5821 | 0.6223 | 0.5964 | 0.6286 | 0.6008 | 0.6318 | 0.6099 | **0.6414** | 0.5805 | 0.6006 |
| $s = 50$ | 0.3383 | 0.3698 | 0.3579 | 0.3757 | 0.3512 | 0.3735 | 0.3706 | **0.3846** | 0.3418 | 0.3448 |
| $s = 100$ | 0.3136 | 0.3135 | 0.3279 | 0.3279 | 0.3232 | 0.3235 | 0.3390 | **0.3395** | 0.2785 | 0.2768 |

'None' indicates neither the head nor the body utilizes our aggregation method (i.e., this is FedAvg); 'Head' means that only the head uses our proposed method (based on DRL); 'Body' means weighted aggregation for the body and simple averaging for the heads; 'Both' is our proposed method (pFedRLLA). 'DRL' means that the aggregation weights generated by the DRL agent are applied to the entire networks.

**Better initialization.** We conducted experiments with variable local training loads of 1 and 5 epochs, and the experimental results are shown in Table 3. We observe that pFedRLLA exhibits a more pronounced advantage under the le = 1 setting, performing even better than some algorithms under the le = 5 setting. This supports its efficiency under limited local work, and can be interpreted to its adaptivity (it generates personalized aggregation weights for each client, which are dynamically adjusted based on client data characteristics). This results in faster convergence and improved accuracy. In contradistinction, algorithms that rely on a single global model for personalization require additional training to capture the unique representative patterns present in the local data, which leads to slower convergence.

Table 3: Experiments with variable local training load (le) measured in epochs. The accuracy boost obtained by pFedRLLA is more accentuated for lower effort (le = 1). Notably, with 80% less load (i.e., le = 1 for pFedRLLA and le = 5 for other methods) our method has the highest accuracy.

| FL settings | | | Average test accuracy | | | | | |
|---|---|---|---|---|---|---|---|---|
| dataset | dist. | le | FedAvg | FedAvg-FT | APFL | FedBABU | FedALA | pFedRLLA |
| CIFAR-10 | $s = 5$ | 1 | 77.46 | 78.35 | 77.86 | 78.55 | 77.57 | **80.14** (1.59 ↑) |
| | | 5 | 79.43 | 80.26 | 80.10 | 79.17 | 79.04 | **81.76** (1.50 ↑) |
| | $\alpha = 0.1$ | 1 | 78.98 | 85.22 | 85.66 | 85.61 | 84.75 | **88.20** (2.54 ↑) |
| | | 5 | 84.31 | 89.24 | 89.38 | 89.55 | 87.94 | **90.26** (0.71 ↑) |
| CIFAR-100 | $s = 5$ | 1 | 59.60 | 62.43 | 62.28 | 63.63 | 64.94 | **70.60** (5.66 ↑) |
| | | 5 | 70.14 | 72.95 | 72.82 | 73.62 | 70.37 | **76.06** (2.44 ↑) |
| | $\alpha = 0.1$ | 1 | 37.12 | 39.92 | 35.10 | 41.96 | 42.00 | **52.42** (10.4 ↑) |
| | | 5 | 49.78 | 52.87 | 52.77 | 53.07 | 44.51 | **53.79** (0.72 ↑) |

**Convergence speed and communication cost.** We include a comprehensive analysis of the computation and convergence time of our proposed algorithm. The experiments are conducted on the CIFAR-10 dataset, with a total of 500 training rounds and a fixed learning rate of 0.01. The detailed results are summarized in Table 4. Regarding computation time, FedAvg exhibits the shortest, while FedPHP demonstrates the longest duration. pFedRLLA incurs an inherent overhead introduced by the MHAE and DRL agent components (which are run on the server side, i.e., no additional burden on the user devices). However, this is remedied by a significantly improved convergence rate. In specific, the accuracy of pFedRLLA is stabilized (to a value that is higher than all baselines) in less overall time: the reduction of 20.5% in total time is deemed significant and jus-

Table 4: The computation time and convergence time for all baselines. Computation time denotes the total time required for training 500 rounds, whereas convergence time refers to the duration for an algorithm to reach a stable state.

| | Computation time | | Convergence time | |
|---|---|---|---|---|
| | Total time | Time/iter. | Time | Accuracy |
| FedAvg | 1095.11 | 2.19 | 870.61 | 0.794 |
| FedProx | 1191.65 | 2.38 | 915.86 | 0.795 |
| Ditto | 1634.04 | 3.27 | 1368.34 | 0.800 |
| APFL | 1438.62 | 2.88 | 1101.75 | 0.801 |
| FedBABU | 1346.77 | 2.69 | 1021.07 | 0.791 |
| FedFomo | 1159.67 | 2.32 | 997.13 | 0.768 |
| FedAMP | 1188.62 | 2.37 | 963.93 | 0.647 |
| FedPHP | 1754.37 | 3.51 | 1291.61 | 0.797 |
| pFedRLLA | 1673.37 | 3.35 | 692.64 | 0.817 |

tifies and compensates for the overhead on the server side. For more analysis of the convergence behavior, we provide supplementary experiments in the Appendix F.5.

## 5 Conclusion

The paper proposed a novel layer-wise aggregation method for personalized federated learning to address data heterogeneity. pFedRLLA leverages different aggregation methods for different layers of neural networks, allowing to learn a better representation while ensuring model personalization. Through ablation experiments, we demonstrated the advantages of the two aggregation methods and confirmed that considering both methods simultaneously can yield a synergistic effect. An extensive experimental analysis illustrated *higher accuracy in less time*.

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

## A    APPENDIX

The supplementary material is organized as follows. Section 1 introduces the datasets used in this paper. Section 2 shows the design of the DRL agent and the details of its training. Section 3 presents additional experiments to assess the performance of the algorithm.

## B    EXPERIMENTAL SETUP

### B.1    DATASETS

We have open-sourced our code on GitHub (`https://anonymous.4open.science/r/pFedRLLA-7BEF`). We adopted the official public codebase of Jianqing Zhang et al. (`https://github.com/TsingZ0/PFL-Non-IID`) to implement our method and other baselines in PyTorch V.1.8.1. For baselines, we have adjusted their hyperparameters to ensure that their best performance is shown in our experiments. The setting of the hyperparameters can be found in the shell script in our code.

In this paper, we evaluate our algorithm on four datasets, CIFAR-10, CIFAR-100, MNIST* and Omniglot. The first two datasets are very common and we will not go into too much detail here. For the latter two datasets, we treat them as heterogeneous scenarios that differ from the first two. The MNIST* dataset is a hybrid dataset constructed from three datasets: MNIST, MNIST-M, and USPS. All three datasets contain digits from 0-9. MNIST consists of handwritten digits from the census, while USPS is a set of digits automatically scanned from envelopes by the US Postal Service. MNIST-M is a dataset that combines MNIST digits with random color patches from the BSDS500 dataset. During the experiment, we drew samples from three distinct datasets. Specifically, we sampled 40% of clients from the MNIST dataset, another 40% of clients from the USPS dataset, and the remaining 20% of clients from the MNIST-M dataset. This sampling strategy ensured that different clients were assigned the same task but from different domains, thereby making the scenario more realistic.

The Omniglot dataset contains 1,623 different handwritten characters from 50 different alphabets. Each of these characters was drawn online via Amazon's Mechanical Turk by 20 different users. It is one of the most naturally separated datasets representing real-world scenarios.

### B.2    HYPERPARAMETERS

For the baselines employed in this work, we conducted an optimization of their hyperparameters. Here, we present the specific configurations utilized in our experiments. For the FedAvg-FT approach, we performed a 5-epoch fine-tuning process. For Ditto, we set $\lambda=0.5$ which is the regularization term control parameter. For APFL, the initial value of $\alpha$ was set to 1, which was then adaptively adjusted based on the algorithm's recommendations. For FedRep, we set the number of local epochs for both the head and body to 5. For FedAMP, $\lambda$ was set to $5 \times 10^{-7}$ and $\alpha_k$ was set to $5 \times 10^{-3}$. For FedPHP, both $\lambda$ and $\mu$ were set to 0.1. For FedFomo, the server only transmits $M$ (=5) client models to each client in every round. Last, for FedALA, the ALA module selected all layers and employed 80% of the local training data for training purposes, while the weight learning rate was set to 1. For more detailed parameter settings, please refer to our shared code.

## C    THE DESIGN OF THE DRL AGENT

We present the deep deterministic policy gradient (DDPG) (Lillicrap et al., 2015) algorithm as the basis for our DRL agent. DDPG is constructed based on the Actor-Critic framework, where the Actor network represents the policy $\mu$, and the Critic network represents the value function $Q$. The Actor network takes the environmental state $s$ as input and outputs the action that maximizes the value $Q$ under the current state, thus forming a deterministic policy. This network directly performs gradient descent on the value function $Q$. The critic network takes the environmental state $s$ and the action $a$ generated by the Actor network as input and outputs the fitted $Q$ value. Notably, DDPG employs the technique of Soft Target Update, which involves maintaining two sets of Actor and

Critic networks: the source network and the target network. This approach aims to enhance the stability of the learning process.

In the training of DDPG, we treat the learning task as a one-step decision task, where each weight aggregation is considered as one task. One-step tasks typically exhibit a more prominent reward signal, which facilitates the agent's understanding of when to receive rewards or penalties. To avoid defining it as a multi-objective task (since we need to aggregate weights for each client), we design the state representation analogously. We consider the clients selected in the previous round and the client for which the aggregated model is currently being generated as part of the state. We place the target client at the end to indicate the objective for each decision, aiming to train the DRL agent to allocate higher aggregation weights to clients similar to the target client.

Regarding the selection of the algorithm for DRL agent, various methods and architectures could be explored. However, we did not try alternative approaches to the one presented because we believe the obtained results demonstrate the merits of the proposed solution, while it is not the focus of this paper to compare RL algorithms in the DRL agent block.

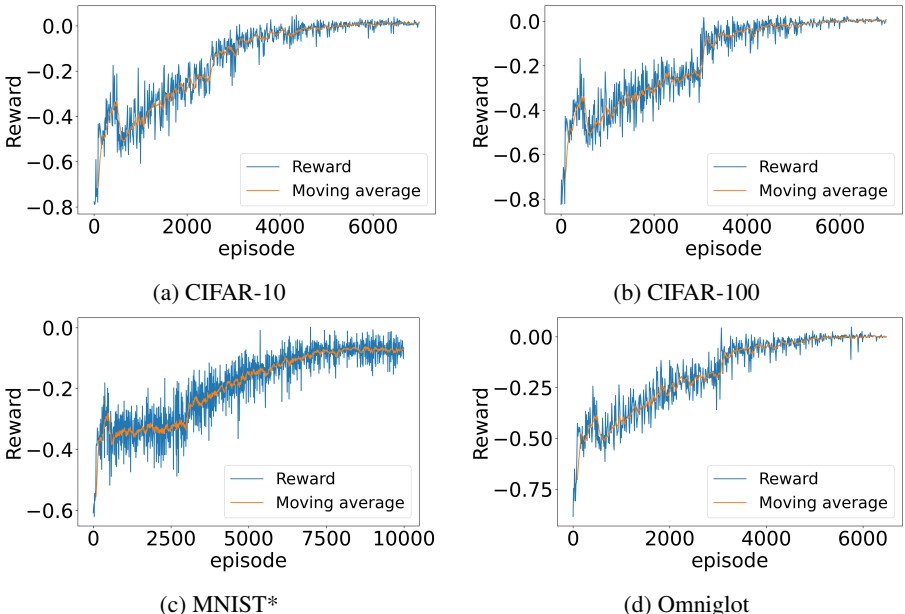

(a) CIFAR-10

(b) CIFAR-100

(c) MNIST*

(d) Omniglot

Figure 2: Training progress of the DRL agent on four datasets. The specific experimental settings are discussed in the first paragraph of appendix D. The moving average is a smoothing technique applied to rewards, accomplished by computing the average of a sequence of consecutive subsequences. It can be inspected that the training process is relatively smooth and stabilizes after 6,000 episodes (equal to 600 communication rounds in our experimental setting).

## D  THE DETAILS OF PRE-TRAIN

In this work, both the DRL agent and the multi-head auto-encoder are pre-trained. It should be noted that pre-training uses a single scenario (in terms of the data held by the users), while testing is done across distinct heterogeneity scenarios. During testing, the DRL/MHAE is further fine-tuned (detailed in Appendix E) *simultaneously* with the PFL process (i.e., without any additional cost in terms of iterations of the algorithm). In specific the pre-training settings are as follows: a) CIFAR-10 wich $s=4$ (different than those tested, which are $s \in \{2, 5\}$). b) CIFAR-100 with $s=10$ (different than those tested, which are $s \in \{5, 20\}$). c) Omniglot data are randomly discarded based on label, to ensure its heterogeneity is different from the test environment. d) for MNIST*, the user ratio across the three datasets is adjusted for both pre-training and testing. Specifically, for the pretraining stage, a ratio of 30% MNIST, 30% USPS, and 40% MNIST-M is used, while for the testing stage, the ratio is adjusted to 40% MNIST, 40% USPS, and 20% MNIST-M. In the following, we give more details

about the pre-training process of auto-encoder and DRL agent on CIFAR-10, respectively (other cases are similar).

**Pre-training of the multi-head auto-encoder (MHAE)**: In each iteration, the Multi-Head Autoencoder (MHAE) is employed to compute a novel embedding (the encoder output) for each selected user. To train the MHAE, the decoder is utilized to generate a reconstructed output, and subsequently, the mean square error (MSE) between the input and reconstructed output is computed as the loss function for updating the MHAE.

**Pre-training of the DRL agent**: The settings used for pre-training are the same as for the MHAE. In each round, we use DRL agent to generate actions (aggregation weight) for each selected user based on the current state; after performing the action, we calculate the reward and the new state. This process forms a (state, new_state, weight, reward) tuple, and we store these tuples to the replay buffer. The replay buffer is continuously updated with new experiences, and the training process keeps cycling through sampling from the buffer. It is worth noting that our reward is directly linked to accuracy. Therefore, in the pre-training process, in order to mitigate the influence of the body and ensure that any accuracy improvements observed in the validation set are solely attributed to the head, we restrict the body to local training only, without any communication with the server. The head does personalized aggregation using the DRL agent.

In Fig. 2, we illustrate the training process of DDPG on four datasets, where each episode represents a decision-making process, and the reward reflects the accuracy of the resulting aggregated model in that particular aggregation.

# E  THE DETAILS OF FINE-TUNING

Our DRL agent is pre-trained. And then it is further fine-tuned simultaneously with the federated process (and the same holds true for the auto-encoder). During each iteration of the fine-tuning process, we will sample 32 data instances from the replay buffer to fine-tune the DRL agent every 10 rounds. As for the multi-head auto-encoder (MHAE), we conduct fine-tuning every 10 rounds. The mean square error (MSE) between the input and output (the decoder output) serves as the loss function for updating the MHAE. In the remaining rounds, we solely focus on using the encoder to obtain the embedding.

In our method, the additional overhead on the server side mainly comes from the DRL agent and MHAE (other operations like updating storage and computing weights based on data volumes have minimum overhead). We demonstrate the costs in the following setting: CIFAR-10 $s$=5, 100 users, global_round=500, local_epoch=5, batch_size=64. In this experiment, we calculated the average time by dividing the total time by 500, which encompasses the local training time (2.37s), DRL time (0.16s), and MHAE time (0.94s). The latter two times are composed of inference time and fine-tuning time. A detailed analysis is presented below:

- The total cost at the server is just 46% of the cost of local training: 85% for MHAE (70% for inference and 30% for fine-tuning), 15% for DRL agent (12% for inference and 3% for fine-tuning).

The above analysis shows that compared with local training time, the computation costs caused by the DRL agent are minimal. The MHAE incurs a significant cost, but it is not a core component. We can use other efficient dimension-reduction methods. In addition, in a non-simulated environment, fine-tuning of the DRL agent and MHAE can be parallel-processed, i.e., when the server is idle (selected clients do local training, and the server is waiting for updated models).

# F  ADDITIONAL EXPERIMENTS

## F.1  DISCUSSION OF REWARD HYPERPARAMETERS

When designing the reward function, we conducted individual experiments for each sub-reward to ensure their effectiveness. To address this, we introduce ablation experiments using $r_1$, $r_2$, and $r_3$ to investigate the impact of each sub-reward (detailed in Table 5). From this, it can be seen that if any coefficient for the three reward components $r_1$, $r_2$, $r_3$, equals 0, this leads to the lowest accuracy;

this corroborates using all three. In addition, when varying a single weight, further gains over default choices (base) can be obtained for a specific setting (CIFAR-10 $s$=5 was used here) in the case of $\beta_1, \beta_3$. For fair reporting of results, we adopt a common setting $\{1,1,2\}$ in all experiments (for all datasets and heterogeneity levels).

Table 5: Ablation experiment of superparameter $\beta$ in reward. The first row indicates the hyperparameters manipulated in the experiment. The second row presents the specific values of the parameter combinations. The last row denotes the average test accuracy. These experiments are conducted at CIFAR-10 $s$=5.

| var. | $\beta_1$ | | | $\beta_2$ | | | $\beta_3$ | | | base |
|---|---|---|---|---|---|---|---|---|---|---|
| $\beta_1 - \beta_2 - \beta_3$ | 0-1-2 | 5-1-2 | 10-1-2 | 1-0-2 | 1-5-2 | 1-10-2 | 1-1-0 | 1-1-5 | 1-1-10 | 1-1-2 |
| Acc | 69.22 | 70.95 | 70.85 | 69.28 | 69.38 | 69.34 | 69.45 | 71.14 | 70.95 | 70.84 |

## F.2 Discussion of pretrained DRL agent

In the preceding description, we pre-train our DRL agent on a dataset and apply it to various heterogeneous scenarios on this dataset. We emphasize that our DRL task is exclusively associated with the employed model in the FL task, utilizing the embedding of model parameters as the state, thereby remaining independent of the FL task. Consequently, we further investigate a *new scenario* where pre-training is conducted on a different dataset than the one used for testing (pre-train on CIFAR-10, test on other datasets). The results presented in Table 6 demonstrate that our algorithm continues to perform effectively under the new scenario. While the experimental results may be comparatively inferior to previous cases, it is essential to note that our algorithm still exhibits advantages when compared with the best baselines in these particular scenarios.

Table 6: Average Accuracy on CIFAR-100, Omniglot, MNIST* datasets. We use the DRL agent pre-trained on CIFAR-10 $s$=2, and then use it on different dataset. 'new' represents pFedRLLA under the new scenario (mentioned in F.2), while "previous" refers to pFedRLLA using the previous method. "best baseline" indicates the highest accuracy achieved among other baseline methods across different experimental settings.

| | CIFAR-100 | | Omniglot | MNIST* |
|---|---|---|---|---|
| | $s = 5$ | $\alpha = 0.1$ | - | - |
| new | 74.53 | 53.01 | 47.74 | 73.90 |
| previous | **76.06** | **53.79** | **48.43** | **74.58** |
| best baseline | 73.62 | 53.21 | 45.36 | 73.88 |

## F.3 Analysis of aggregation weights.

To demonstrate that our algorithm can assign higher weights to similar clients (without any prior knowledge of similarities), we designed and conducted an additional set of experiments. We consider a set of nine clients with full participation at each round and record the weight values of each client on the target participant during the training process. The results are listed in Fig. 3 with the data partitioning on the left, the heatmap of weights of all clients in the middle, and the weights for a single client (client 1) on the right, for better visualization.

From the visualization on the left, we may divide the commonalities between the clients into three types: (1) similar distributions, i.e., users that have the same class (for example user 1 and user 2 in the CIFAR-10 example); (2) different distributions, with the same class (i.e., user 1 and user 4 in the CIFAR-10 example); (3) different distributions, without common class (i.e., user 1 and user 8 in the CIFAR-10 example). In the heatmaps, the illustration is column-wise, i.e., the weights at the $i-$th column are the contributions of all clients to target client $i$. We observe that the brightness of the diagonal is very high, which means that the self-weight is relatively large. Combined with the setting of data distribution, we can also find that the more similar the clients are, the larger the weight of contribution during aggregation. When there are fewer similar clients, the self-weight is

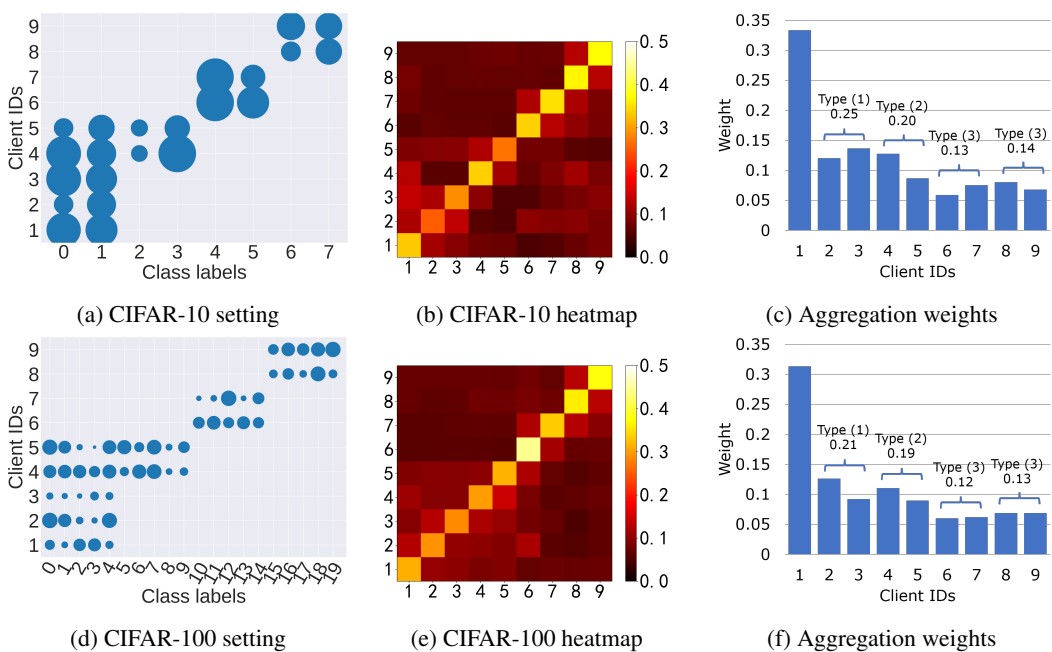

Figure 3: Analysis of aggregation weights. The first column is the visualization of the non-IID data distribution of the designed experiment on CIFAR-10 and CIFAR-100. The second column is a heatmap of aggregated weights for all clients at the end of training. With client 1 as the target, the row of data corresponding to client 1 in the heatmap is the aggregation weight for other clients. The third column is the aggregation weight of client 1.

very large, but at the same time, the algorithm still keeps learning new knowledge from others. For instance, the aggregation weights for client 1 are depicted in Fig. 3c, revealing that clients belonging to type 1 exhibit relatively higher aggregation weights. Type 2 clients represent the second highest weights, while clients 6-9 (type 3) possess dissimilar labels to client 1, leading to comparatively lower weights being assigned to them.

### F.4    MORE DETAILS OF THE ABLATION STUDY

In this section, we present additional results based on the ablation study outlined in the main paper. For specific details regarding these settings, please refer to Table 7. Our primary focus here is to evaluate the impact of two distinct aggregation methods when applied to different components of the model. Our findings demonstrate that when employing weighted aggregation solely on the body, the model exhibits higher test accuracy prior to the other two configurations in 'Weighted Aggregation'. Furthermore, this approach demonstrates further improvement after undergoing personalized training. Conversely, when personalized aggregation is exclusively applied to the head, superior performance is observed compared to the other two settings. These results serve to provide supporting evidence for our choices. In light of these outcomes, our proposed algorithm incorporates weighted aggregation and personalized aggregation for the body and head, respectively. This particular configuration yields the most favorable performance among all tested settings.

### F.5    THE CURVE OF TEST ACCURACY

In this section, we depict the accuracy curves and the number of rounds required for each algorithm to reach a prescribed target accuracy. Detailed results are presented in Fig. 4. We observe that pFedRLLA achieves the best performance among the compared baselines, achieving a 40% communication savings compared with the best baseline in (a), (b), and (c). Based on the results presented in Table 1, it is evident that FedALA exhibits superior test accuracy compared to the majority of baseline methods. However, a closer examination of Fig. 4 reveals that FedALA generally demonstrates comparatively slower progress at the initial stages when compared to other algorithms across various scenarios.

Table 7: Ablation study. This is a supplement to the ablation experiments in the main text, which records the test accuracy before and after personalization. In each sub-table, the second line indicates the aggregation method used, and the third line specifies the particular part of the model where the aggregation method is applied.

| | Before Personalization (BP) | | | | | | |
|---|---|---|---|---|---|---|---|
| | Weighted Aggregation | | | Personalized Aggregation | | | pFedRLLA |
| | body | head | both | body | head | both | |
| $s = 10$ | 0.5931 | 0.5253 | 0.5890 | 0.5947 | 0.6008 | 0.5805 | 0.6099 |
| $s = 50$ | 0.3706 | 0.2482 | 0.3516 | 0.3766 | 0.3512 | 0.3418 | 0.3706 |
| $s = 100$ | 0.3279 | 0.1873 | 0.3001 | 0.3279 | 0.3232 | 0.2785 | 0.3390 |
| | After Personalization (AP) | | | | | | |
| | Weighted Aggregation | | | Personalized Aggregation | | | pFedRLLA |
| | body | head | both | body | head | both | |
| $s = 10$ | 0.6282 | 0.5253 | 0.6072 | 0.6304 | 0.6318 | 0.6006 | 0.6414 |
| $s = 50$ | 0.3744 | 0.2482 | 0.3552 | 0.3807 | 0.3735 | 0.3448 | 0.3846 |
| $s = 100$ | 0.3279 | 0.1873 | 0.2987 | 0.3279 | 0.3235 | 0.2768 | 0.3395 |

## F.6 INCREASING SELECTION RATIO.

Fig. 5 demonstrates multiple experiments with variable participant selection ratio ($c \in \{0.1, 0.25, 0.5\}$). It can be seen that pFedRLLA is very robust to the choice of $c$, which is in support of both the communication and training efficiency of the method (the lower the participation, the lower the communication and local training). Besides, we can also observe that the performance of FedAvg deteriorates as the selection ratio increases. This is in line with the intuition because FedAvg does not employ personalized learning, and a large number of heterogeneous clients is confusing to the algorithm. FedBABU, FedPHP, and FedALA exhibit significant performance fluctuations under certain data distributions (especially in Fig. 5b). Comparing FedAvg-FT and FedBABU, we find that FedBABU outperforms FedAvg-FT in the CIFAR-100 experiments, but the opposite is true under the CIFAR-10 experimental settings. This indicates that FedBABU has limitations in certain scenarios. In contrast, our algorithm, pFedRLLA, demonstrates consistent and best performance in various heterogeneous scenarios and demonstrates robustness to changes in the selection ratio.

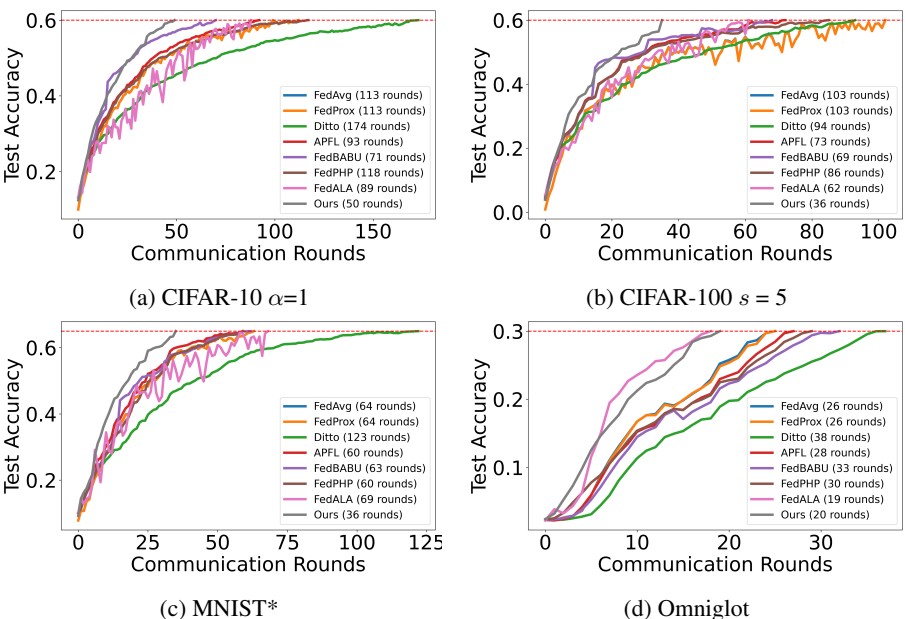

(a) CIFAR-10 $\alpha$=1      (b) CIFAR-100 $s = 5$

(c) MNIST*      (d) Omniglot

Figure 4: Test accuracy trajectories. pFedRLLA portrays the fastest convergence speed. For example, on CIFAR-10, our algorithm needed 50 rounds to achieve 0.6 accuracy, while the best of the other baseline required 89 rounds.

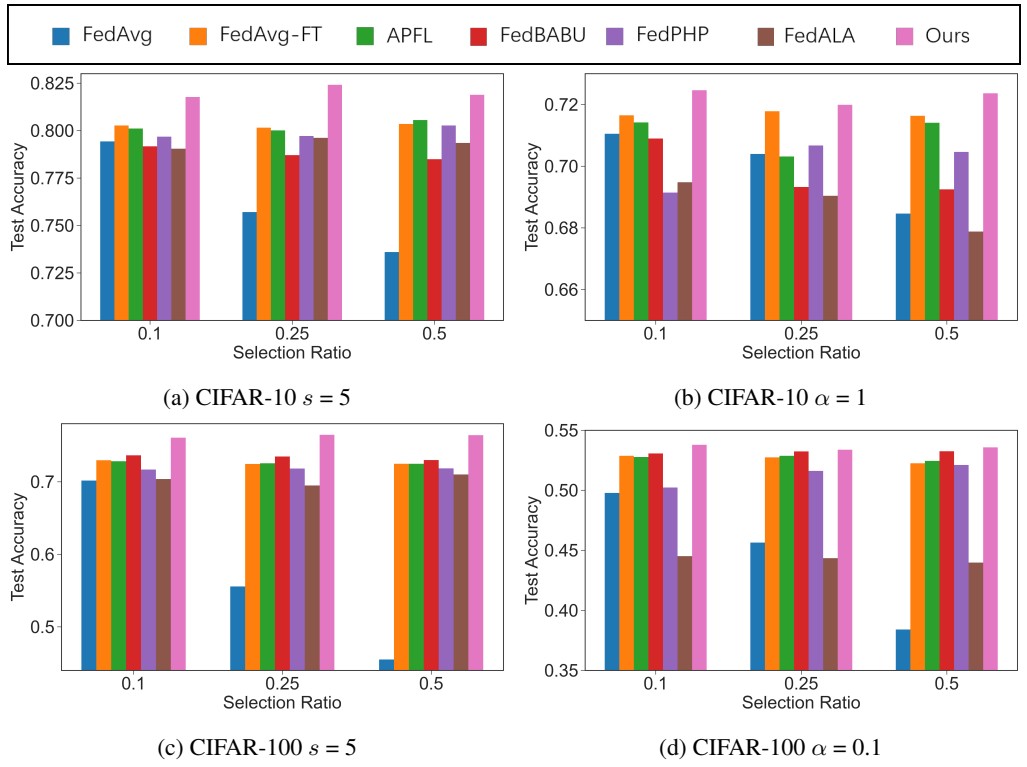

Figure 5: As the selection ratio grows, our algorithm can still maintain an advantage over other baselines, showing demonstrating robustness.

To verify the effectiveness of our dimension reduction method, we design the following experiment on CIFAR-10: 100 clients are divided into five groups according to the distribution of their local data. Each group of clients has data with two labels, and different groups of clients have different labels. We run our algorithm for 100 rounds. Before the end of the algorithm, we use the PCA and MHAE we designed to extract the features of the model parameters. Fig. 6 illustrates a comparative visualization of MHAE vs. PCA. It can be observed that the MHAE achieves a denser intra-group distribution and a clearer inter-group boundary, indicating its superiority over the PCA method in our context.

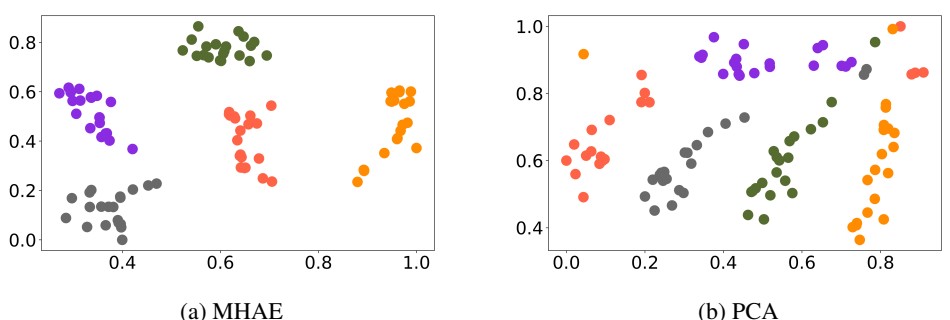

Figure 6: Visualization of the embedding obtained by using the multi-head auto-encoder. (b) is the visualization result of embedding using PCA. Clients are divided into 5 groups based on data distribution, each of which is distinguished using a different color.

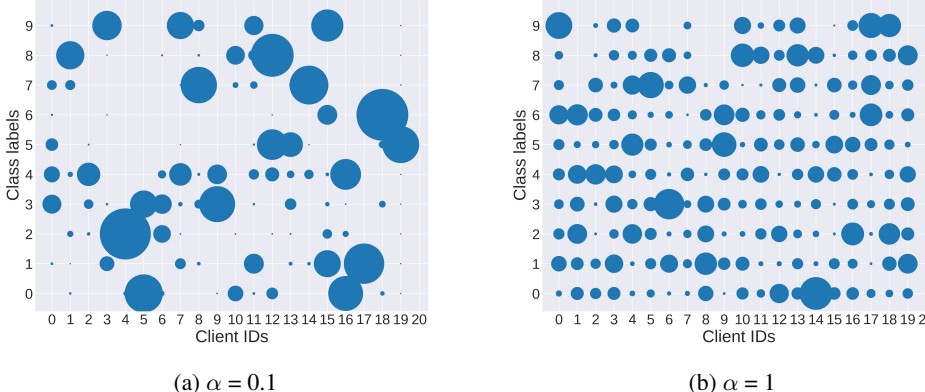

(a) $\alpha = 0.1$                    (b) $\alpha = 1$

Figure 7: Visualization of the Dirichlet non-IID scenario. In this figure, we employed the Dirichlet data partitioning method to divide the CIFAR-10 dataset into 20 subsets. This choice was made due to the impracticality of visualizing the dataset when divided into 100 subsets. The size of the balls represents the data volume. It is worth highlighting that the observed heterogeneity is even more pronounced when the dataset is partitioned into 100 clients.

## G    THE EXPERIMENTS WITHOUT PRE-TRAINING

We conducted experiments without pre-training, and the specific details of the experiment are outlined as follows:

- Firstly, we modified the experiment settings to require each user to maintain two local models. One model, denoted as $w_p$, is utilized for training and uploading and is updated through layer-wise aggregation. The second model, denoted as $w_{DRL}$, is employed for validation testing, and its test accuracy is used to calculate the reward. Notably, $w_{DRL}$ does not undergo training; instead, it updates its head once $w_p$ is updated (by copying from $w_p$). The update timing for the body of $w_{DRL}$ depends on a signal received from the server, as described in detail below.

- Secondly, the training process for both the DRL agent and MHAE can be likened to fine-tuning, as mentioned in Appendix E. The main distinction lies in the initial 50 rounds of federated learning, during which the DRL agent randomly generates aggregation weights (sampled from the interval [0, 1] and subsequently normalized). MHAE, on the other hand, undergoes training in every round, and after 50 rounds, it undergoes fine-tuning every 10 rounds. Additionally, whenever the DRL agent is updated, a signal is sent to all users to prompt them to update the body of $w_{DRL}$ (by copying from the latest $w_p$).

- Lastly, the remaining experiment settings for training are consistent with those outlined in Section 4.1 (100 users, 500 rounds, learning rate=0.01, and selection ratio=0.1).

The experimental settings are 100 users, 500 rounds, learning rate=0.01, and selection ratio=0.1, the same as the previous setting in the paper (section 4.1, page 6). From the Table 8, we can observe that the new algorithm without pre-training can achieve better results than the baseline. However, the attained accuracy is, of course, lower than before because of zero-shot training for DRL/MHAE. We further explore the slowdown in the Fig. 8. This figure illustrates the learning process of the algorithms. Fig. 8a shows that the new algorithm initially has a slower convergence speed, and it reaches 90% of the previous algorithm in 100 rounds and 95% of the previous algorithm in 200 rounds. As the process continues, the difference between the two algorithms becomes smaller and smaller (1% after 500 rounds).

Table 8: Without pre-train. The experimental settings are 100 users, 500 rounds, learning rate=0.01, and selection ratio=0.1. In this figure, we present the average accuracy achieved by the three methods. 'best baseline' refers to the highest performance observed among the baseline approaches. 'previous' denotes our algorithm utilizing a pre-trained DRL (Deep Reinforcement Learning) agent, while 'new' represents our algorithm without any pre-training.

|  | CIFAR-10 | | CIFAR-100 | | Omniglot | MNIST* |
|---|---|---|---|---|---|---|
|  | $s$=2 | $s$=5 | $s$=5 | $s$=20 | - | - |
| best baseline | 89.80 | 80.26 | 73.62 | 48.04 | 45.36 | 73.88 |
| previous | **90.81** | **81.76** | **76.06** | **51.15** | **48.43** | **74.58** |
| new | 89.93 | 81.63 | 75.12 | 51.03 | 46.06 | 74.24 |

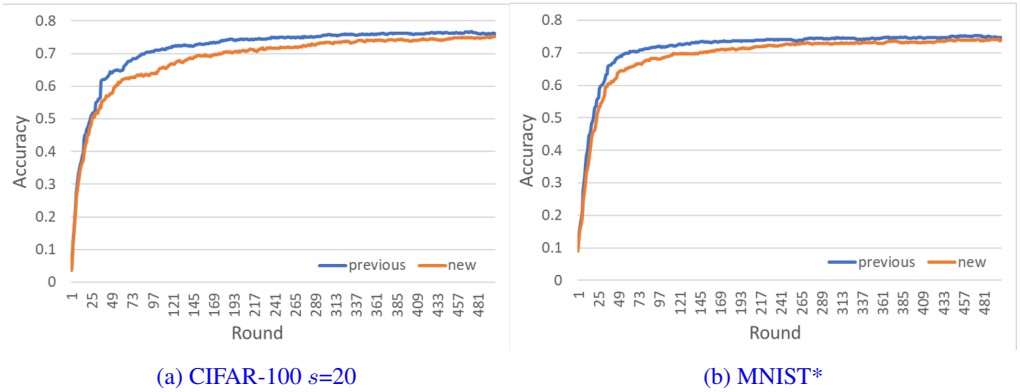

(a) CIFAR-100 $s$=20

(b) MNIST*

Figure 8: The convergence process of our algorithm. We conducted a comparison between its convergence with pre-training (previous) and without pre-training (new). In CIFAR-10 $s$=20, it can be found that without pre-training, the gap (defined as: (prev Acc. - new Acc.) / prev. Acc.) is about 10% after 100 rounds, 5% after 200 rounds, and 1% after 500 rounds.