# OpenReview forum: "Reinforcement Learning-based Layer-wise Aggregation for Personalized Federated Learning"
_ICLR.cc/2024/Conference — Submitted to ICLR 2024_

### Official Review · Reviewer_M4kH · 2023-10-25

**Soundness:** 3 good
**Presentation:** 3 good
**Contribution:** 2 fair
**Rating:** 6
**Confidence:** 3

**Summary:**

This paper proposes a reinforcement learning-based layerwise aggregation method (pFedRLLA) that applies different mechanisms for different neural network layers. pFedRLLA leverages different aggregation methods for different layers of neural networks, allowing to learn a better representation while ensuring model personalization. The authors conduct extensive experiments to validate the proposed methods which reach state-of-the-art performance.

**Strengths:**

- The paper investigates the personalized federated learning problem which is a hot and important topic. Utilizing reinforcement strategy in PFL is also interesting.

- The paper is well-organized and easy to understand. The figures are clear for illustrations.

- The authors conduct extensive experiments to validate the proposed methods which reach state-of-the-art performance.

**Weaknesses:**

- The authors adopt $\beta$ as the hyper parameters on different rewards. However, how to properly choose $\beta$? In my opinion, it should be discussed in the paper.

- Some similar paper should be discussed in the main paper, e.g., [1]

- Some information is missing in the main paper, e.g. the name of Alg.2 is missing.

Ref:
[1] Zhu, Suxia, Tianyu Liu, and Guanglu Sun. "Layer-Wise Personalized Federated Learning with Hypernetwork." Neural Processing Letters (2023): 1-15.

**Questions:**

Please refer to the weaknesses above

---

> ### Author Response · Authors · 2023-11-21
> **Response to Reviewer M4kH**
>
> # Answer for Q1
> Thank you very much for pointing this out: it is our negligence that we had not explained the tuning of $\beta_i$ in the paper. This was carried through experimentation, and we have adopted default parameters {1,1,2} (listed before (1a) on page 6 in the revised paper) for all experiments.
> Additionally,  we have included Table 5 in Appendix F.1 (page 14) to show the ablation study for choosing the weights for the three reward components. We also include here for your convenience:
>
> |var.||$\beta_1$|||$\beta_2$|||$\beta_3$||base|
> |-|-|-|-|-|-|-|-|-|-|-|
> |$\beta_1$-$\beta_2$-$\beta_3$|0-1-2|5-1-2|10-1-2|1-0-2|1-5-2|1-10-2|1-1-0|1-1-5|1-1-10|1-1-2|
> |acc|69.22|70.95|70.85|69.28|69.38|69.34|69.54|71.14|70.95|70.84|
>
> This is the process of pre-training the DRL agent using different hyperparameter combinations (detailed in Appendix D). The experimental setting is to make `CIFAR-10 s=5`.
>
> From this, it can be seen that if any coefficient for the three reward components $r_1, r_2, r_3$, equals 0, this leads to the lowest accuracy; this corroborates using all three. In addition, when varying a single weight, further gains over default choices (base) can be obtained for a specific setting (`CIFAR-10 s=5` was used here) in the case of $\beta_1,\beta_3$. For fair reporting of results, we adopt a common setting {1,1,2} in all experiments (for all datasets and heterogeneity levels).
>
> # Answer for Q2
> Thank you very much for suggesting this reference. We have listed and discussed this paper in the last paragraph of section 2 (page 3). We also include here for ease of your inspection:
>
> pFedLHN [1] employs layer-wise modules with parameter-sharing mechanisms in the hypernetwork to create personalized models for each client. The hypernetwork generates these personalized models by utilizing layer embeddings uploaded by the clients themselves and does not rely on learning from other clients. Moreover, uploading these layer embeddings to the server introduces additional communication.
>
> **Ref**:
> [1] Zhu, Suxia, Tianyu Liu, and Guanglu Sun. "Layer-Wise Personalized Federated Learning with Hypernetwork." Neural Processing Letters (2023): 1-15.
>
> # Answer for Q3
> Thank you for your useful feedback on our paper. We have named Alg. 2 as "Layer-wise aggregation", and the default parameters for $\beta_i$ are now listed in the main paper (on page 6, before (1a)). The details for pre-training the DRL agent have been included in Appendix D (page 13).
>
> We sincerely appreciate the time you took to review our paper and provide us with helpful suggestions!

---

> > ### Comment · Reviewer_M4kH · 2023-11-22
> >
> > Thank you for your response. It effectively addresses my concerns. I will maintain my positive rating.

---

### Official Review · Reviewer_qRiv · 2023-10-30

**Soundness:** 1 poor
**Presentation:** 3 good
**Contribution:** 1 poor
**Rating:** 3
**Confidence:** 4

**Summary:**

The paper provides a personalized federated learning method based on reinforcement learning, which aggregates the local models in a layer-wise manner. In the method, the body part of the model  (i.e., shared knowledge) is aggregated according to the local data size. Also, the head part of the model (i.e., personalized knowledge) is aggregated by using the personalized aggregation weights that are generated by deep reinforcement learning.

**Strengths:**

The proposed algorithm uses multi-head autoencoder for dimension reduction instead of PCA, which makes it more practical. The source code of the paper is provided.

**Weaknesses:**

For personalized federated learning, plenty of works have been studied using a variety of learning methods such as knowledge distillation, meta learning, transfer learning, etc., and including reinforcement learning. Considering this, the concept of the paper is not sound and novel, and its contribution seems marginal compared with the conventional federated learning methods with reinforcement learning and multi-armed bandits.

**Questions:**

1. The main concepts of the paper such as layer-wise aggregation, the use of reinforcement learning for model weighting, and dimension reduction have been already considered in federated learning. Please clarify the contribution of the paper compared with the related works, currently, it seems that the contribution is marginal.

2. In a similar context, it is not clear why reinforcement learning is suitable for personalization of federated learning. Learning another deep neural network for DRL agent may incur significant costs. Please justify the rationale of using DRL agent for personalized federated learning.

3. Typically, a lot of experiences is required to learn a DRL policy. In federated learning, is there an enough number of rounds that the DRL policy to be converged? Or, should the DRL policy be trained in advance?

4. How is the multi-head autoencoder trained when using the proposed algorithm? Is it should be trained in advance? This should be clarified in the paper.

5. If the DRL policy or autoencoder should be trained in advance, it should be clarified how the dataset for training them can be obtained before applying federated learning.

6. For a limited number of rounds, it may be doubted that considering accuracy-related rewards (i.e., $r_1$ and $r_2$) will be effective or not. Providing the ablation study of $r_1$, $r_2$, $r_3$ would be helpful to understand the effect of the weights according to the reward structure.

7. It seems that the reward in Eq. (1d) is structured so that the weights closer to the similarity vector have the smaller reward. Hence, it encourages  the weight of the similar client become smaller. But, in my understanding, the concept of the proposed algorithm should encourage the weight of the similar client become larger for personalization.

8. It seems that the weights $p_k$'s from the DDPG should be constrained as to be 1 in sum. How is this realized in the algorithm?

---

> ### Author Response · Authors · 2023-11-21
> **Response to Reviewer qRiv (1/2)**
>
> # Answer for Q1
> Thank you for this valuable suggestion. You are right that the novelty of the proposed algorithm does not lie on the tools invoked, but rather on the succinct design of a new PFL framework with emphasis on scalability at the server end and no overhead on the user end.
>
> In specific, this is accomplished by a) adopting a single DRL agent (for all participants), b) using a compound reward that balances both cross-user similarities and local accuracy improvement, and c) obtaining model embeddings with a single MHAE block to alleviate costs and boost the performance of the DRL weight inference. In addition, we have demonstrated through numerous experiments that a simple and efficient layer-wise approach (body and head) is effective in this framework.
>
> In our introduction section, we have provided a clearer explanation after the contributions (page 2), of the novelty of our work. We would like to point out that these choices significantly differ from existing methods described in the literature (i.e., rather than simply combining choices from various personalized algorithms, we have utilized standard tools to develop a novel solution for PFL), as elaborated next:
>
> a. **Reinforcement Learning**: first, we adopt *a single* DRL agent (for all participants), enabling higher scalability. Second, we use a compound reward that balances accuracy and model similarities. The design of rewards is explicit, which allows a faster convergence. Third, our DRL task is formulated as a single-step task, which not only provides a more pronounced reward signal but also simplifies the collection of training data.
>
> b. **Dimension reduction**: we employ a multi-head autoencoder (MHAE) (i.e., that MHAE tries to get relationships between different layers of the same network). Besides, adopting a single MHAE block alleviates costs and boosts the performance of DRL weight inference.
>
> c. **Layer-wise aggregation**: previous works [3, 4, 5] only considered similarity of user models. In comparison, we considered not only similarity but also the data volume based on the simple observation [1,2] that the body aims to learn domain-specific features while the head task-specific features. Doing this allows us to get a better feature extractor (improve model generalization and capture data diversity and complexity) and a more customized classifier (give us a faster convergence speed) efficiently.
>
> **Refs**:
>
> [1] Decoupling representation and classifier for long-tailed recognition.
>
> [2] The Devil is the Classifier: Investigating long tail relation classification with decoupling
> analysis.
>
> [3] Layer-Wise Personalized Federated Learning with Hypernetwork.
>
> [4] Personalized Federated Learning with Feature Alignment and Classifier Collaboration.
>
> [5] Layer-wised Model Aggregation for Personalized Federated Learning.
>
> # Answer for Q2
> Thank you for raising an important point about the use of DRL in the proposed algorithm. In the revised paper we have listed all the details in Appendix D (page 13).
>
> Our DRL agent is pre-trained. Then it is further finetuned simultaneously with the federated process (and the same holds true for the auto-encoder). During each iteration of the fine-tuning process, we will sample 32 data instances from the replay buffer to finetune the DRL agent every 10 rounds. For the MHAE, we conduct fine-tuning every 10 rounds. We list the details of pre-train in the appendix D (page 13) and fine-tuning in the appendix E (page 14).
>
> In our method, the additional overhead on the server side mainly comes from the DRL agent and MHAE (other operations like updating storage and computing weights based on data volumes have minimum overhead). We demonstrate the costs in the following setting: `CIFAR-10 s=5, 100 users, global_round=500, local_epoch=5, batch_size=64`. In this experiment, we calculated the average time by dividing the total time by 500, which encompasses the local training time (2.37s), DRL time (0.16s), and MHAE time (0.94s). The latter two times are composed of inference time and fine-tuning time. A detailed analysis is presented below:
>
> - The total cost at the server is just 46% of the cost of local training: 85% for MHAE (70% for inference and 30% for fine-tuning), 15% for DRL agent (12% for inference and 3% for fine-tuning).
>
> The above analysis shows that compared with local training time, the computation costs caused by the DRL agent are minimal. The MHAE incurs a significant cost, but it is not a core component. We can use other efficient dimension-reduction methods. In addition, in a non-simulated environment, fine-tuning of the DRL agent and MHAE can be parallel-processed, i.e., when the server is idle (selected clients do local training, and the server is waiting for updated models).

---

> ### Author Response · Authors · 2023-11-21
> **Response to Reviewer qRiv (2/2)**
>
> # Answer for Q3 & Q4 & Q5
> Thank you very much for these questions. In all our experiments the DRL agent and the MHAE are pre-trained and fixed for all experiments (**answer for Q3 and Q4**). This was carried in the following setting: For each dataset, we pre-train a DRL agent, which can be used in different heterogeneous scenarios of the corresponding dataset. It is important to note that the pre-training setting differs entirely from the experimental settings employed in our paper, which has been discussed in detail in the revised paper in Appendix D (first paragraph, on page 13). When the pre-trained model is in use, we fine-tune it simultaneously with the federated process, as explained in the detailed response to Q2.
>
> **Answer for Q5**, due to space reasons, we do not show the detailed process of pre-training here. We have discussed this in detail in Appendix D (page 13).
>
> As our DRL approach is solely associated with the model employed (the embedding of model parameters used as the state), it is independent of the task itself. therefore, we further study a scenario where pre-training is carried on a different dataset than the one tested. We have discussed this in detail in Appendix F.2 (page 15).
>
> ||CIFAR-100||Omniglot|MNIST*|
> |-|-|-|-|-|
> ||$s$=5|$\alpha$=0.1|-|-|
> |new|$\underline{74.53}$|53.01|$\underline{47.74}$|$\underline{73.90}$|
> |previous|**76.06**|**53.79**|**48.43**|**74.58**|
> |best baseline|73.62|53.21|45.36|73.88|
>
> # Answer for Q6
> Thank you for your question. You are right that the choices for the reward are worthy of a detailed discussion, which we include in the updated paper in Appendix F.1 (page 14). We also include here for your convenience:
>
> |var.||$\beta_1$|||$\beta_2$|||$\beta_3$||base|
> |-|-|-|-|-|-|-|-|-|-|-|
> |$\beta_1$-$\beta_2$-$\beta_3$|0-1-2|5-1-2|10-1-2|1-0-2|1-5-2|1-10-2|1-1-0|1-1-5|1-1-10|1-1-2|
> |acc|69.22|70.95|70.85|69.28|69.38|69.34|69.54|71.14|70.95|70.84|
>
> This is the process of pre-training the DRL agent using different hyperparameter combinations (detailed in Appendix D). The experimental setting is to make `CIFAR-10 s=5`.
>
> From this, it can be seen that if any coefficient for the three reward components $r_1, r_2, r_3$, equals 0, this leads to the lowest accuracy; this corroborates using all three. In addition, when varying a single weight, further gains over default choices (base) can be obtained for a specific setting (`CIFAR-10 s=5` was used here) in the case of $\beta_1,\beta_3$. For fair reporting of results, we adopt a common setting {1,1,2} in all experiments (for all datasets and heterogeneity levels).
>
> # Answer for Q7
> Thank you for your attentiveness. You are absolutely right: there was a typo in (1d) (- sign missing), for which we apologize and have corrected and explained before (2) in the revised paper (on page 6).
>
> # Answer for Q8
> Thank you for pointing this out, which we should have explained in the  revised paper (on page 6, paragraph 2). We apply a softmax activation function to the final layer of our network, which normalizes the outputs to add up to 1 as follows: $softmax(p_{k}) = \frac{exp(p_k)}{\sum exp(p_k)}$

---

> > ### Comment · Reviewer_qRiv · 2023-11-21
> >
> > I appreciate the great effort the authors have made to respond and make revisions. I believe the paper is quite improved, but the most significant concern is still not addressed.
> >
> > I strongly disagree that the training cost for the DRL agent and MHAE is negligible because they are pretrained. Typically, a pretrained model implies a model that can be used for different datasets/tasks with only simple fine-tuning. However, what the authors did in the experiments can be seen as a preparation stage of the proposed approach, in which the DRL agent and MHAE are trained before applying the proposed approach, rather than building a pretrained model. (Note that if someone tries to apply the proposed approach to a new real-world application, typically, he/she cannot train the DRL agent and MHAE as the authors did because there is no such known scenario of the application). In Appendix F.2, the results of using the DRL agent pretrained with a different dataset are presented, but it does not seem to be as effective. Furthermore, the different architectures of neural networks require different MHAEs for the proposed approach. This makes it difficult to use a pretrained MHAE in a variety of applications.
> >
> > In summary, I still have serious concerns about the cost of the proposed approach, and this still significantly degrades the contributions of the work. Therefore, I maintain my score.

---

> > > ### Author Response · Authors · 2023-11-23
> > >
> > > We would like to thank you once again for your constructive criticism. We acknowledge that the test setting is not fully convincing for practical scenarios (but would like to kindly point out that this is commonplace for the vast majority of FL methods that aim to provide prototype algorithms and lack a real system implementation). Besides,
> > >
> > > - In a real-scenario, pre-training of the DRL agent can be carried offline just like when pre-training the neural network (e.g., by artificially diving pre-training data across a number of users, with an additional overhead that is less than the model training effort). Our design choices facilitate reusability, in that MHAE is applied only on the heads (so that different architectures with common heads dimension can use the same pre-trained DRL/MHAE).
> > >
> > > - We have evaluated this by pre-training on a dataset and testing on another (see Table 6, on page 15), where the content is totally different (e.g., CIFAR-10 has animals, vehicles, etc., while MNIST* and Omniglot have digits).
> > >
> > > - In addition, we have carried new experiments **without pre-training** for DRL/MHAE (see Table 8, page 20, also listed below for your convenience). In this experiment, the training of DRL/MHAE is carried *simultaneously* with the federated process (every 10 rounds). After 130 rounds, our method outperforms all baselines in terms of accuracy, with an additional overhead (on the server side) of 89% of local training effort at the edge: 90% for MHAE (65.7% for inference and 24.3% for training), 10% for DRL agent (5.3% for inference and 4.7% for training).
> > >
> > >   **The new experiments without pre-training**. The experimental settings are `100 users, 500 rounds, learning rate=0.01, and selection ratio=0.1`. More details of these experiments have been included in Appendix G. The experiment results are as follows:
> > >
> > >   |               | CIFAR-10     |              | CIFAR-100    |              | Omniglot     | MNIST*       |
> > >   | ------------- | ------------ | ------------ | ------------ | ------------ | ------------ | ------------ |
> > >   |               | s=2          | s=5          | s=5          | s=20         | -            | -            |
> > >   | best baseline | 89.80        | 80.26        | 73.62        | 48.04        | 45.36        | 73.88        |
> > >   | previous      | **90.81**    | **81.76**    | **76.06**    | **51.15**    | **48.43**    | **74.58**    |
> > >   | new           | $\underline{89.93}$ | $\underline{81.63}$ | $\underline{75.12}$ | $\underline{51.03}$ | $\underline{46.06}$ | $\underline{74.24}$ |
> > >
> > >   The attained accuracy is, of course, lower than before because of zero-shot training for DRL/MHAE. We further explore the slowdown in the Fig .8 in Appendix G (page 20).
> > >
> > >   The primary information of Fig. 8 is as follows: the convergence speed of 'new' is slower than that of 'previous', and the accuracy gap between them is relatively large at the beginning, and then more minor becomes smaller. In specific, without pre-training, the gap (defined as: (prev Acc. - new Acc.) / prev. Acc.) is about 10% after 100 rounds, 5% after 200 rounds, and 1% after 500 rounds.
> > >
> > > - Last but not least, there are two federated learning works based on RL [1, 2], and they also have a pre-train process, the detailed comparison is as follows:
> > >
> > >   - Favor [1] proposes a deep Q-learning mechanism that selects a subset of devices in each communication round to maximize a reward, promoting increased validation accuracy and discouraging excessive communication rounds. In their paper (section EVALUATION, subsection A, page 7), they discuss the training process of their DRL agent. The agent achieves convergence after more than 150 episodes, where each episode starts at the initialization of a federated learning job and ends when the job reaches the target accuracy (representing the entire federated learning convergence process).
> > >   - DearFSAC [2] dynamically assign weights to local models to improve the robustness of FL. In their paper (Fig. 5), they show the learning process of DRL agent. their setting is total episodes T is 800 and each episode contains 50 FL rounds.
> > >
> > > We have also modified the paper accordingly (see Appendix G, on page 19). We would like to thank you once more for your very useful feedback!
> > >
> > > **Refs**:
> > >
> > > [1] Wang, Hao, et al. "Optimizing federated learning on non-iid data with reinforcement learning." IEEE INFOCOM 2020-IEEE Conference on Computer Communications, 2020.
> > >
> > > [2] Huang, Chenghao, et al. "DearFSAC: An Approach to Optimizing Unreliable Federated Learning via Deep Reinforcement Learning." arXiv:2201.12701*, 2022.

---

### Official Review · Reviewer_54Ry · 2023-11-01

**Soundness:** 3 good
**Presentation:** 2 fair
**Contribution:** 2 fair
**Rating:** 6
**Confidence:** 3

**Summary:**

This paper introduces pFedRLLA, a reinforcement learning-based layer-wise aggregation method designed to tackle the challenge of statistical heterogeneity in personalized federated learning. The approach applies different aggregation strategies to different layers of neural networks. The feature extracting layers (body) use weights proportional to the data sizes of the clients, while the heads employ a deep reinforcement learning (DRL) agent to generate personalized aggregation weights. The DRL agent considers compound rewards that take into account both the improvement of validation accuracy and the similarity between clients. Besides, to reduce the state space, a multi-head auto-encoder is utilized to obtain low-dimensional embeddings of user models.

Experimental results on benchmark datasets with varying levels of data heterogeneity demonstrate the effectiveness of the proposed method. It outperforms the baselines in terms of accuracy and convergence speed.

**Strengths:**

1. This paper presents a novel approach that utilizes reinforcement learning (specifically, DDPG) to tackle the task of learning aggregation weights in the classifier heads. This approach seems to offer a fresh perspective.

2. The proposed method surpasses existing approaches in addressing data heterogeneity in personalized federated learning. The results demonstrate higher accuracy achieved in less time.

**Weaknesses:**

1.The authors claim that the main novelty lies in the layer-wise design in section 2 (related work). However, the idea of separating the body and head for aggregation is not entirely novel, as seen in approaches like pFedLA, which even achieves a finer-grained layer-wise aggregation using the power of hypernetworks. In comparison, this paper only roughly separates the body and head, and cannot be considered truly layer-wise like pFedLA. Additionally, the idea of separating aggregation for the body and head has also been proposed in FedPAC, with a more detailed mathematical logic. Besides, the weighted averaging based on data volume for body aggregation is trivial and straightforward.

2. Although the experimental results demonstrate the superior performance of the pFedRLLA framework, there is a lack of theoretical analysis explaining why RL-based methods work better. It seems that the paper provides more intuitive reasoning.

3. In terms of clarity, the paper occasionally fails to clearly differentiate between crucial and supplementary information. For instance, the descriptions accompanying the figures and tables, such as Figure 1 and Table 2, are excessively long, while the corresponding text in the main body appears relatively weaker. The steps mentioned below Figure 1 are relatively concise, but the figure description provides excessive detail, also including redundant content.

**Questions:**

1. In the reward design, the hyperparameter "beta" is not further explained, and it appears that in the code it is simply set as {1, 1, -2}. Regarding the weight allocation for r1, r2, and r3 (the compacts of validation accuracy and similarity), it is unclear if there is a more in-depth consideration. By the way, it would be better to add a negative sign to r3 to maintain consistency with r1 and r2.

2. Will there be any theoretical analysis provided to explain why RL-based methods outperform existing pFed methods? Besides DDPG, have you attempted other RL methods, and why is DDPG the preferred choice?

3. Regarding clarity, there is room for improvement about the issues raised in Weakness3.

---

> ### Author Response · Authors · 2023-11-21
> **Response to Reviewer 54Ry**
>
> # Answer for Weakness1
> Thank you very much for these useful comments. We have re-written the related parts in the introduction (page 2) to explain the novelty in our approach which does not lie on the used tools per se, but rather on the design of the framework (targeting scalability on the server end and no additional burden to the users). To that end, through extensive experiments, we have tried to demonstrate that the simplest possible layer-wise approach (body & head) is efficient and suffices to yield superior accuracy over numerous baselines.
>
> Regarding layer-wise aggregation, pFedLA [3] only considered the similarity of user models. In contrast, we considered not only similarity but also the data volume based on the simple observation [1,2] that the body aims to learn domain-specific features while the head task-specific features. This allows us to get a better feature extractor (improve model generalization and capture data diversity and complexity) and a more customized classifier (a faster convergence speed is attained). In FedPAC [4], the computation and transmission of feature centroids introduce additional computational and communication costs, both on the server and client sides. Moreover, its applicability is limited to scenarios involving label distribution shift. In contrast, our algorithm imposes no additional computational or communication burdens on the client side.
>
> **Refs**:
>
> [1] Decoupling representation and classifier for long-tailed recognition.
>
> [2] The Devil is the Classifier: Investigating long tail relation classification with decoupling analysis.
>
> [3] Personalized Federated Learning with Feature Alignment and Classifier Collaboration.
>
> [4] Layer-wised Model Aggregation for Personalized Federated Learning.
>
> # Answer for Q1
> Thank you very much for your question. First, you are absolutely right: there was a typo in (1d) (- sign missing), for which we apologize and have corrected and explained before (2) in the revised paper.
>
> It was also our negligence that we had not explained the tuning of $\beta_i$ in the paper. This was carried through experimentation, and we have adopted default parameters  {1,1,2} (listed before (1a) on page 6 in the revised paper) for all experiments.
> Additionally,  we have included Table 5 in Appendix F.1 (page 14) to show the ablation study for choosing the weights for the three reward components. We also include here for your convenience:
>
> |var.||$\beta_1$|||$\beta_2$|||$\beta_3$||base|
> |-|-|-|-|-|-|-|-|-|-|-|
> |$\beta_1$-$\beta_2$-$\beta_3$|0-1-2|5-1-2|10-1-2|1-0-2|1-5-2|1-10-2|1-1-0|1-1-5|1-1-10|1-1-2|
> |acc|69.22|70.95|70.85|69.28|69.38|69.34|69.54|71.14|70.95|70.84|
>
> This is the process of pre-training the DRL agent using different hyperparameter combinations (detailed in Appendix D). The experimental setting is to make `CIFAR-10 s=5`.
>
> From this, it can be seen that if any coefficient for the three reward components $r_1, r_2, r_3$, equals 0, this leads to the lowest accuracy; this corroborates using all three. In addition, when varying a single weight, further gains over default choices (base) can be obtained for a specific setting (`CIFAR-10 s=5` was used here) in the case of $\beta_1,\beta_3$. For fair reporting of results, we adopt a common setting {1,1,2} in all experiments (for all datasets and heterogeneity levels).
>
> # Answer for Q1 & Weakness2
> **Answer 2.1**: Unfortunately, our proposed method relies solely on data-driven techniques for which it is especially challenging to provide theoretical guarantees (in the absence of a commonly adopted model). Instead, we have resorted to extensive experimentation over heterogeneous settings to corroborate the efficacy and robustness of the proposed solution.  This line of research has proliferated in the FL literature [1, 2, 3].
>
> **Answer 2.2**: You are absolutely right that other RL methods (e.g., PPO, TRPO) can be used in the proposed framework. We opted for DDPG (the reason is clarified in the first paragraph of section 3.2.1) and have not tested with any other alternatives. We appreciate the reviewer's understanding that this would not be feasible in the rebuttal period, however, we explicate the choice in the revised paper in Appendix C (last paragraph, on page 13).
>
> **Refs**:
>
> [1] FedFormer: Contextual Federation with Attention in Reinforcement Learning.
>
> [2] Layer-Wise Personalized Federated Learning with Hypernetwork.
>
> [3] Clustered Federated Learning: Model-Agnostic Distributed Multitask Optimization Under Privacy Constraints.
>
> # Answer for Q3 and Weakness 3
> Thank you for your suggestion. We fully agree that the descriptions were excessively long (in an attempt to summarize the content in one place for easier inspection). In the revised paper, we have shortened the caption of Figure 1 (page 4) by 40% by removing the unnecessary content. Regarding Table 2 (page 8), we also shortened the description by 50%.

---

### Author Response · Authors · 2023-11-21
**General Response**

Dear Reviewers,

We would like to express our sincere gratitude for taking the time to review our paper to provide constructive suggestions. Your feedback has been invaluable in improving the content and presentation of the paper. We have addressed all your comments in the revised paper (edits are shown in blue) and accompanying individual responses. In the following, we provide a summary of the key changes.

### **Main changes**:

- **Introduction**: We have included a paragraph following the contributions to clarify the novelty of our proposed solution. The novelty does not solely rely on the tools employed, but rather on the framework design itself, which prioritizes scalability on the server end while ensuring no additional burden on the users.

- **Related work**: We included a comparison with pFedLHN (as suggested by Reviewer M4kH), a related algorithm that shares similarities with both pFedLA and our proposed approach.

- **Section 3.2.1**:

  - There was a typo in (1d) (- sign missing), we have corrected it (pointed out by Reviewers 54Ry, qRiv).

  - The aggregation weight generated by DRL should sum to 1. We answered how this is realized in the algorithm (as suggested by Reviewer qRiv).

- **Experiments**:

  - We have added experiments on reward hyperparameter analysis. We have included the results in Table 5 in Appendix F.1 (page 14).

  - We have conducted additional experiments involving a DRL agent pre-trained on CIFAR-10 and applied to a different dataset. The results have been included in Table 6, which can be found in Appendix F.2 (page 15).

- **Writing**:

  - We have shortened the captions of Fig. 1 and Table 2 to eliminate redundant details (as pointed out by Reviewer 54Ry).
  - We have added detailed descriptions of how DRL agents are pre-trained (suggested by Reviewer qRiv), which have been included in Appendix F.1 (page 14). In addition to this, we also explain the details of fine-tuning in Appendix F.2 (page 15).

Finally, we would like to thank again all reviewers for your insightful feedback. We sincerely welcome additional comments and questions for further clarification during the rebuttal period.
Sincerely,

---

### Meta-Review · Area_Chair_djKo · 2023-12-09

**Metareview:**

The submission proposes a reinforcement learning based approach to personalized federated learning, with the assumption. that certain layers in a network will represent general knowledge, while others will represent the personalization.  A DRL agent is used to generate "personalized" aggregation weights.  The review received mixed borderline positive and more strongly negative review scores, with reviewers being positive about empirical results, but being less positive about motivation and practicality of the reinforcement learning approach.

**Justification For Why Not Higher Score:**

The submission proposes another personalized federated learning scheme based on a reinforcement learning strategy for aggregation policy.  Reviewers raised questions about the justification for this approach, and I myself question why the use of reinforcement learning is proposed in place of strategies that consider priors over the relationship between networks, such as using variational Bayesian inference as in Zhang et al. (ICML, 2022) and other related work.  It would be useful if the submission had included a better interpretation of what reinforcement learning had learned, and if this could rather be encoded in a prior for a more principled unified probabilistic framework.

**Justification For Why Not Lower Score:**

N/A

---

### Decision · Program_Chairs · 2024-01-16

Reject